# The Bidirectional Relationship of NPY and Mitochondria in Energy Balance Regulation

**DOI:** 10.3390/biomedicines11020446

**Published:** 2023-02-03

**Authors:** Diana Sousa, Eduardo Lopes, Daniela Rosendo-Silva, Paulo Matafome

**Affiliations:** 1Coimbra Institute for Clinical and Biomedical Research (iCBR) and Institute of Physiology, Faculty of Medicine, University of Coimbra, 3000-548 Coimbra, Portugal; 2Center for Innovative Biomedicine and Biotechnology (CIBB), University of Coimbra, 3004-504 Coimbra, Portugal; 3Clinical Academic Center of Coimbra (CACC), 3004-561 Coimbra, Portugal; 4Polytechnic Institute of Coimbra, Coimbra Health School, Rua 5 de Outubro—S. Martinho do Bispo, 3046-854 Coimbra, Portugal

**Keywords:** NPY, mitochondria, energy balance, metabolic disease

## Abstract

Energy balance is regulated by several hormones and peptides, and neuropeptide Y is one of the most crucial in feeding and energy expenditure control. NPY is regulated by a series of peripheral nervous and humoral signals that are responsive to nutrient sensing, but its role in the energy balance is also intricately related to the energetic status, namely mitochondrial function. During fasting, mitochondrial dynamics and activity are activated in orexigenic neurons, increasing the levels of neuropeptide Y. By acting on the sympathetic nervous system, neuropeptide Y modulates thermogenesis and lipolysis, while in the peripheral sites, it triggers adipogenesis and lipogenesis instead. Moreover, both central and peripheral neuropeptide Y reduces mitochondrial activity by decreasing oxidative phosphorylation proteins and other mediators important to the uptake of fatty acids into the mitochondrial matrix, inhibiting lipid oxidation and energy expenditure. Dysregulation of the neuropeptide Y system, as occurs in metabolic diseases like obesity, may lead to mitochondrial dysfunction and, consequently, to oxidative stress and to the white adipose tissue inflammatory environment, contributing to the development of a metabolically unhealthy profile. This review focuses on the interconnection between mitochondrial function and dynamics with central and peripheral neuropeptide Y actions and discusses possible therapeutical modulations of the neuropeptide Y system as an anti-obesity tool.

## 1. Introduction

Obesity has been rising over the years, and it is associated with chronic energy imbalance and a higher risk of other metabolic diseases such as metabolic syndrome, prediabetes, non-alcoholic fatty liver disease, and type 2 diabetes. The intricate process of energy storage/expenditure and food intake/satiety regulation is orchestrated by the gut hormones adipokines, and neuropeptides. In particular, the neuropeptide Y (NPY), is a potent orexigenic peptide pointed out as an obesogenic factor [1]. Indeed, patients with obesity have higher levels of NPY which may promote higher food intake and lower energy expenditure [1]. Moreover, exposure to high-fat (HF) and high-sugar diets in rodents increases NPY levels and sensitivity in the hypothalamus, contributing to weight gain and fat accumulation [2]. Thus, changes in NPY levels may precede obesity and be one of the main driving factors for its development.

Mitochondria are the energy producers of the cell, and part of the ATP production results from lipid oxidation. Mitochondrial dysfunction is common in metabolic diseases, namely obesity [3], in which mitochondrial dynamics and mitochondrial gene expression are altered, leading to lower ATP production [3]. NPY/Agouti-related peptide (AgRP) neurons are highly dependent on FA β-oxidation, suggesting that NPY release may be conditioned by mitochondrial function and energy status [4]. Some studies have suggested that NPY is involved in mitochondrial dynamics, not only at the central level but also in peripheral organs [5]. Thus, the crosstalk between NPY and mitochondria should be explored further as a new perspective of treatment for energy imbalance conditions. Nevertheless, some questions are still unclear regarding NPY’s impact on mitochondria function. With this review, we intend to address the role of mitochondria in the regulation of the NPY system and understand the influence of energy balance dysregulation caused by NPY on energy production and mitochondrial biogenesis.

## 2. NPY as the Master Regulator of Energy Balance

NPY is the most powerful neuropeptide in controlling appetite at the central level [6]. However, NPY is more than an appetite regulator since it also exerts actions on energy storage/expenditure by emitting signaling through the autonomic nervous system (ANS) and by NPY receptors (NPYRs)-binding in peripheral cells such as adipocytes. The ANS regulates anabolic and catabolic processes, controlling energy balance [7,8,9]. Obesity is highly associated with an imbalance between sympathetic and parasympathetic tones. As an important peptide in the ANS, NPY signaling dysregulation is associated with energy balance perturbances and, therefore, with obesity. Indeed, several studies showed that in individuals with obesity, NPY serum levels are higher compared to lean subjects [10]. NPY is altered even between different obesity phenotypes: metabolically healthy obesity (MHO), in which the subjects present insulin sensitivity and normoglycemia, and metabolically unhealthy obesity (MUO), in which the subjects, besides having different fat distribution, also demonstrate insulin resistance, hyperglycemia, dyslipidemia, and a more inflammatory environment [11]. Subjects with MUO have higher serum levels of NPY compared with individuals with MHO, demonstrating the crucial role of NPY in metabolic diseases [12]. Deletion of the *Npy* gene attenuates weight gain in mice fed an HF diet, showing the involvement of NPY in obesity development [13]. Moreover, NPY is also known for its adipogenic and hyperplasic effects on white adipose tissue (WAT). The NPY system is formed by three native forms of peptides, NPY (NPY_1-36_), peptide YY (PYY), and pancreatic polypeptide (PP), and by their cleaved forms. Mature NPY is a 36 amino acids peptide and upon mRNA pre-proNPY transduction, post-translational modifications (PTM) may occur. Several enzymes may perform these PTMs, such as dipeptidyl peptidase 4 (DPP4), responsible for the cleavage of NPY_1-36_ into NPY_3-36_ [14]. Other NPY fragments were already found, such as NPY_13-36_, NPY_18-36_, and NPY_22-36_, although their roles have been more associated with the cardiovascular system [15,16]. Recently, it was found that the pre-proNPY mRNA can also originate another NPY fragment (NPY_17-36_) by alternative splicing, suspected to act primarily on mitochondria, which function will be discussed in Section 4. Besides all the different peptides, six NPYRs are known, all coupled to an inhibitory G protein [17]. The NPYRs are widespread, being in brain centers and peripheral organs but their precise functions are not yet fully disclosed and might differ according to cell type. The NPY receptor-1 (NPY1R) and NPY receptor-5 (NPY5R) are associated with anabolic processes [6,18]. In the hypothalamus, NPY1R and NPY5R have a similar outcome, stimulating food intake [19]. Despite the high levels of NPY5R mRNA in adipose tissue, its peripheral role is not yet established, whereas NPY1R has a peripheral obesogenic effect [19,20,21,22]. The role of NPY receptor-2 (NPY2R) is cell-specific, playing an autoinhibitory role in the hypothalamus, whereas it promotes adipogenesis in the WAT [19]. Despite being present in humans, little is known regarding NPY receptor-3 (NPYR3) downstream effects. The NPY receptor-4 (NPY4R) has been characterized by its anti-obesity action and is related to satiety [23]. NPY receptor-6 (NPY6R) is widely expressed in mice and rabbits, but it was not found in primates (humans, chimpanzees, gorillas, and tamarins) [24,25]. The affinity for each NPY receptor is dependent on the amino acid composition and folding of the ligand [26,27]. For instance, NPY1R requires a ligand with a full N-terminus, but it tolerates amino acid substitutions in the middle and on the C-terminal of NPY peptides [26,27,28]. The integrity of the C-terminal is essential to NPY2R binding, whereas dissociation of the first two amino acids enhances the affinity for NPY2R and therefore NPY_3-36_ and PYY_3-36_ present higher affinity for NPY2R [27]. The NPY5R is the less specified, binding to 1-36, 2-36, and 3-36 NPY-like peptides [27]. The NPY4R is also known as a PP receptor (PPR) presenting higher affinity to PP than NPY peptides due to a hairpin-like fold only observed in PP [26,27]. NPY is produced in NPY/AgRP neurons of the arcuate nucleus (ARC) of the hypothalamus, in the dorsomedial nucleus of the hypothalamus (DMH), in extrahypothalamic neuronal cells (nucleus accumbens, hippocampus), in adipocytes, and islet immature cells [29,30,31,32]. Besides NPY’s role in feeding, this potent obesogenic peptide decreases insulin sensitivity and glucose metabolism in peripheral tissues such as the liver, brown adipose tissue (BAT), heart, and skeletal muscle [33,34]. Moreover, NPY can act as a growth factor in islet cells and regulate cardiac function, indicating that, although not yet entirely understood, the NPY system is extremely important in several physiologic activities, most of them related to anabolic function [17,35].

### 2.1. NPY Regulation and Feeding Regulation—Ghrelin and Leptin Take the Control

In ARC, both anorexigenic and orexigenic neurons have projections to the paraventricular nucleus of the hypothalamus (PVH), known as the community center, and to the satiety and feeding centers (the ventromedial nucleus of the hypothalamus (VMH) and the lateral hypothalamic area (LHA), respectively), which are all connected through neuronal signals [36]. Upon anorexigenic and orexigenic peptides release, their receptors are activated, and the signal is received by LHA, VMH, and PVH. For instance, the effect of NPY on feeding control seems to be mediated by NPY1R accompanied by NPY5R [19]. In LHA and PVH, NPY/AgRP neurons stimulate food intake while suppressing satiety in VMH, possibly by inhibiting neurons expressing steroidogenic factor 1 (SF1) that are responsible for inducing satiety through the paraventricular thalamus (PVT) [37]. However, the LHA area is also highly related to hedonic feeding [38].

NPY/AgRP neurons are activated by ghrelin in low glucose level conditions, as well as by mechanical signals emitted by the gut through the vagus nerve [39]. In postprandial situations, high glucose levels induce AMP-activated protein kinase (AMPK) inhibition, leading to a decline in firing NPY/AgRP neurons, since free fatty acids (FFA) constitute their main source of energy [4,40,41,42]. Proopiomelanocortin (POMC) neurons use glucose as an energy source [43]. POMC neurons release anorexigenic peptides, namely α-melanocyte-stimulating hormone (α-MSH), which activates its receptor, the melanocortin 4 receptor (MC4R), inhibiting food intake and increasing energy expenditure [36,44]. The relation between NPY and POMC neurons is well known (Figure 1A). POMC and NPY/AgRP neurons inhibit each other, controlling energy expenditure and appetite. During fasting conditions, the neurotransmitter gamma-aminobutyric acid (GABA) is co-released with NPY, inhibiting POMC neurons [36,45]. NPY inhibits MC4R expression in the hypothalamus, attenuating food intake suppression [36]. Furthermore, the α-MSH action on food intake in PVH is inhibited by AgRP, which also competes for the MC4R [46,47]. On the postprandial state besides the action of the autoinhibitory receptor NPY2R (activated by cleaved NPY_3-36_ and PYY_3-36_) on NPY/AgRP neurons, POMC neurons release α-MSH that binds to MC4R on NPY/AgRP neurons, inhibiting the firing of orexigenic neurons and suppressing food intake induced by NPY. However, this system is more complex since there are several subpopulations of POMC neurons [48,49]. Regarding energy balance, we can classify two of them: (1) one is responsible for the inhibition of food intake independently of energy expenditure through MC4R activation in PVH [POMC neurons expressing the glucagon-like peptide-1 receptor (GLP-1R)—POMC^Glp−1r^]; (2) the other subpopulation stimulates energy expenditure by stimulating MC4R activity in the dorsal vagal complex (DCV) and the intermediolateral nucleus (IML) (POMC neurons expressing leptin—POMC^Lepr^) [36,45,48,49].

NPY/Agouti-related peptide (AgRP) neuron activity is regulated by several peripheral signals. During fasting, the hunger hormone ghrelin is the most well-known orexigenic hormone. As mentioned above, ghrelin plays a critical role in the activation of NPY/AgRP neurons. Ghrelin is an orexigenic hormone released in the fasting state, promoting food intake and adiposity [50]. Ghrelin acts on the hypothalamus, by binding to growth hormone secretagogue receptor 1α (GHS-R1α) in NPY/AgRP neurons [51]. The pre-proghrelin mRNA, produced by P/D1 cells in humans, is converted into mature ghrelin in its two forms: acyl-ghrelin which can bind to GHS-R1α [50,51]; and des-acyl ghrelin which is the most common form found in circulation, who’s receptor has not yet been identified [50]. The ghrelin-O-acyl transferase (GOAT) is responsible for ghrelin acylation, using readily available fatty acids from the diet [52,53,54,55]. However, its release occurs during fasting periods, which is thought to induce food intake and reduce energy expenditure through a GHSR1α-independent mechanism [52,53]. As occurs with NPY levels, in patients with obesity, the ratio of acyl/des-acyl ghrelin duplicates in fasting conditions [56]. For many years, des-acyl ghrelin was thought of as a hormone without metabolic activity. However, it has been suggested that during fasting, this form may play a role as an opponent of acyl-ghrelin regarding glucose homeostasis [51,57]. Acyl-ghrelin inhibits insulin release and increases glucose plasma levels; however, des-acyl ghrelin inhibits this anti-incretin effect of acylated ghrelin [51,57]. The adipokine leptin is an already well-established inhibitor of ghrelin action. Leptin is an anorexigenic hormone produced in proportion to the amount of adipose tissue [58,59]. Within the hypothalamus, the leptin receptor decreases NPY/AgRP neurons firing, reducing food intake while stimulating POMC neurons to trigger energy expenditure [45,58,60]. Leptin only acts on POMC neurons associated with energy expenditure—the POMC^lep^ neurons—not stimulating the POMC neurons related to satiety (POMC^GLP−1R^ neurons) [36,45]. However, leptin promotes satiety by suppressing NPY/AgRP neurons and, therefore, decreasing the inhibitory tone to POMC^GLP−1R^ neurons.

Obesity is highly associated with leptin and ghrelin resistance [59]. Hyperleptinaemia is a common feature among subjects with obesity [59,61,62]. Leptin resistance is characterized by the incapacity of leptin to cross the blood–brain barrier (BBB) and to activate the leptin receptor, due to the inability to bind to the circulating receptor (LepRe), which is essential for the transportation of leptin into the brain [59,61,62]. This inability to sensitize leptin contributes to overfeeding and weight gain. Consequently, ghrelin and leptin resistance can disrupt the NPY system, contributing to energy balance dysregulation.

Regarding satiety, several other hormones and peptides can regulate the process by acting in both NPY/AgRP and POMC neurons. For instance, during the postprandial state, glucagon-like peptide-1 (GLP-1), insulin, cholecystokinin (CCK), PP, PYY, and leptin are released, directly acting on the hypothalamus or activating the sympathetic nervous system (SNS) in vagus afferent nerve fibers which project to nucleus tractus solitarius (NTS) [39,63,64]. The 36 amino acid peptide GLP-1 released from L-cells binds to its receptor GLP-1R on POMC neurons leading to their depolarization and, therefore, increasing α-MSH-mediated NPY/AgRP neurons inhibition [65]. The direct effect of GLP-1 on NPY/AgRP neurons is still unclear. A recent study by Ruska et al. showed the presence of GLP-1R in NPY/AgRP neurons. Nevertheless, the authors did not show the hyperpolarization of NPY/AgRP neurons upon GLP-1R activity [65]. Insulin and leptin bind to their receptors on POMC neurons leading to depolarization dependent on phosphoinositide 3-kinases (PI3K) activity [63,64]. In addition to stimulating anorexigenic action, insulin inhibits NPY/AgRP neurons by inducing hyperpolarization. In turn, leptin acts through the downregulation of forkhead box protein O1 (*Foxo1*), which directly stimulates *Pomc* gene transcription, leading to α-MSH release and inhibition of NPY neurons [63,64]. CCK is a hormone released by I cells known for promoting satiety [39,63,66]. Indeed, CCK binds to CCK1R on POMC neurons stimulating α-MSH release and in CCK1R vagus nerve afferent fibers that project to NTS promoting satiety [32,67,68]. PP is released by PP cells and can also promote satiety. PP may inhibit food intake by acting directly on the hypothalamus and in the brainstem due to the presence of NPY4R in these areas [69,70]. However, the major effect of PP on satiety is dependent on PP binding to NPY4R on the vagus nerve afferent fibers [23,70,71,72]. The PYY_3-36_, produced by L-cells, is the most common form found in CNS, interacting mainly with NPY2R, and playing an important role in NPY/AgRP neurons inhibition [39]. Therefore, PYY_3-36_ promotes satiety by inhibiting NPY/AgRP neurons, suppressing the POMC neuron inhibition and, therefore, allowing a greater release of α-MSH [39]. Moreover, PYY delays gastric emptying [39] which activates NPY/AgRP neurons through mechanoreceptors. PYY together with leptin is considered a long-term endocrine factor in satiety, whereas GLP-1, CCK, and insulin have short-term effects [39].

### 2.2. NPY Regulates Energy Balance through Energy Expenditure Inhibition

#### 2.2.1. Central NPY Effects

NPY also plays a role in reducing energy expenditure by modulating the SNS activity and by activating NPYRs in peripheral organs. The pathways regulated by NPY regarding feeding control and energy expenditure are distinct from each other. Within the hypothalamus, anorexigenic and orexigenic signaling are emitted to the IML of the spinal cord (central SNS) which sends neuronal signals to the sympathetic ganglia [73]. From this ganglia chain arises the efferent neurons that surround peripheral organs (peripheral SNS) [73]. The VMH is known as the center of SNS [74,75]. Thus, stimulation of VMH neurons by anorexigenic peptides increases the activity of efferent sympathetic nerves [36,74,76]. NPY inhibits VMH neurons, reducing sympathetic innervation activity and therefore, energy expenditure [77,78,79], in opposition to the melanocortin system that stimulates VMH [80]. Indeed, several studies showed a decrease in thermogenesis upon NPY intracerebroventricular (icv) injection. Icv NPY administration reduces uncoupling protein 1 (UCP1) in BAT mitochondria, weakening the thermogenesis process (Figure 2A) [81]. So, at the hypothalamic level, NPY induces sympathetic innervation inhibition, thus decreasing thermogenesis in BAT, suggesting that mitochondrial function decreases upon NPY action on the VMH center [82,83,84,85,86]. The melanocortin system has two subpopulations of neurons, the ones that project signals to PVH, regulating feeding, and the others, in the VMH, in which the MC4R activation increases energy expenditure. However, the PVH center also regulates BAT thermogenesis, but not through a response mediated by α-MSH [47]. Instead, oxytocin acts as an anorexigenic neurotransmitter released from PVH upon GLP-1R activation, stimulating thermogenesis in BAT by SNS stimuli [39,87].

The WAT is also surrounded by SNS fibers. Similar to BAT, the SNS-mediated NPY effect in WAT is an obesogenic stimulus that stimulates fat accretion. For instance, lipoprotein lipase (LPL), an essential enzyme for lipogenesis, is increased in WAT upon NPY icv injection [81]. Furthermore, on *Mcr4* knockout mice, the inhibition of NPY/AgRP neurons enhances the WAT browning by increasing UCP1, indicating that NPY inhibits WAT browning not by decreasing α-MSH but by directly inducing central inhibition of SNS innervation to WAT (Figure 2B) [86]. The NPYRs responsible for regulating adiposity by central sympathetic innervation are still unknown, however, given that deletion of *Npy1r* in the hypothalamus does not alter adiposity [88] and that NPY2R plays an autoinhibitory role. Lipolytic and lipogenic processes may be controlled by NPY5R since its agonism is associated with lower energy expenditure in rats [89,90]. Indeed, hypothalamic ablation of *Npy2r* induced weight gain in mice, suggesting a dual effect of hypothalamic NPY on body weight regulation, stimulating adiposity via NPY5R, and preventing it by acting on NPY2R instead [91]. Furthermore, NPY does not inhibit lipolysis through innervation unless this metabolic process has been previously stimulated [92]. So, NPY decreases lipolysis through peripheral SNS; however, it is still unclear whether this is an effect mediated through the NPYRs or whether it is a consequence of NPY action on WAT adrenergic receptors which are responsible for triggering lipolysis [92].

#### 2.2.2. Peripheral NPY Effects

NPY release in peripheral tissues appears to be dependent on insulin levels [30,93], suggesting that NPY action in the periphery occurs in the postprandial state in which insulin levels are high. Although BAT expresses NPY receptors (NPY1R and NPY5R), Kohei Shimada et al. proved that NPY does not exert any effect on BAT thermogenesis through local activation of the NPYRs [83]. NPY treatment on brown adipocytes in vitro did not alter the oxygen consumption rate, an indicator of thermogenesis, whereas treatment with norepinephrine (NE), a classical thermogenesis stimulator [94], increased this rate [83]. Moreover, NE increased ERK activation and cyclic AMP (cAMP) levels, upstream of UCP1 [93,95], while treatment with both NE and NPY did not induce alterations in NE action. This indicates that NPY prevents energy expenditure in BAT, exclusively through central activation of NPY/AgRP neurons, decreasing peripheral sympathetic tone to BAT [83].

On WAT, NPY has an adipogenic/lipogenic effect via peripheral sympathetic innervation and directly in adipocytes. For instance, NPY2R activation in sympathetic nerves drives preadipocyte proliferation, and adipogenesis and stimulates *Npy2r* expression in adipocytes [96]. Total deletion of *Npy* in mice increased lipolytic proteins levels such as sirtuin 1 (SIRT1) and phospho-hormone-sensitive lipase (HSL) (ser563), whereas downregulated lipogenic genes such as *peroxisome proliferator-activated receptor γ* (*Pparγ*), *CCAAT-enhancer-binding proteins* (*C*/*EBP*), *adipocyte protein 2* (*ap2*), and *diacylglycerol O-Acyltransferase 1* (*DGAT1*) in WAT (Figure 2B) [97]. In cultured 3T3-L1 adipocytes, NPY increased lipogenesis (PPARγ, C/EBP, ap2, and DGAT1) while inhibiting lipolysis (decreasing pHSL (ser563)), via NPY1R activation, suggesting a direct action of NPY in adipocytes to mediate this metabolic response [18,19,31,97]. Peripheral *Npy1r* knockdown in mice prevented diet-induced obesity (DIO), exclusively due to decreased adiposity, rather than alterations in food intake or excretion and physical activity, which can be the result of augmented lipid oxidation capacity [98]. NPY2R also plays an obesogenic role in WAT [19]. An NPY2R antagonist curbed the proliferation of preadipocytes, and endothelial cells stimulated with NPY, compromising adipogenesis and angiogenesis [96]. Moreover, NPY leads to fat accretion in B6.V-*Lep*^ob^/J mice, but this effect is lost with NPY2R antagonist [96]. NPY5R’s direct effect on WAT is not well known. Patients with obesity have higher NPY5R levels in visceral and subcutaneous fat [20]. Insulin resistance and body weight are correlated with NPY5R visceral levels [20]. Nevertheless, the downstream pathways triggered upon NPY5R activation on WAT are unknown. Mashiko et al. demonstrated that NPY5R antagonist administration via the oral route increases uncoupling protein 3 (UCP3) and β-3 adrenergic receptor (β_3_AR) proteins that are related to lipolysis [99]. Indeed, it is possible that these alterations evoked by NPY5R modulation may occur via neuronal mechanisms like central sympathetic innervation, or through a direct hormone action on NPY5R in adipocytes [99].

To summarize, NPY action on WAT depends on neuronal and endocrine signals (Table 1). On the hypothalamus, NPY5R appears to be responsible for adiposity, whereas the direct obesogenic action of NPY in WAT is mediated by NPY1R, inhibiting lipolysis while enhancing adipogenesis and lipogenesis. Moreover, despite the anti-obesity role of NPY2R at the hypothalamic level, in WAT, NPY2R stimulates adipogenesis and angiogenesis, a process crucial to healthy WAT expansion.

The liver is one of the most important organs in lipid homeostasis. During fasting, WAT releases FFA into the plasma that later undergoes metabolization in the liver via β-oxidation to produce energy [100,101]. Therefore, the uptake of these non-esterified fatty acids (NEFA) by the liver is dependent on the WAT lipolysis rate, which is inhibited by NPY (Figure 2B) [100,101]. However, NPY also plays a direct role in hepatic lipid oxidation despite the downstream pathways of NPY1R and NPY2R still remaining unknown [19]. The peripheral *Npy1r* abolition increases carnitine palmitoyltransferase I (CPT-1) levels in the liver, which is a protein required for the mitochondrial β-oxidation of long-chain fatty acids (Figure 2C) [98,100]. This suggests that at the hepatic level, NPY1R downregulates *CPT-1* expression in the outer mitochondrial membrane (OMM). Moreover, it is well established that peroxisome proliferator-activated receptor α (PPAR-α) stimulates *CPT-1* expression; however, it is not yet known whether this transcription factor is a target of NPY [100].

The skeletal muscle also plays an important role in lipid homeostasis. The deletion of *Npy1r* on the periphery has the same outcome regarding CPT-1 on skeletal muscle; however, it also increases oxidative phosphorylation (OXPHOS) and peroxisome proliferator-activated receptor-γ coactivator-1α (Pgc-1α) levels in a sex-dependent manner (Figure 2D) [98]. Thus, the NPY system is a crucial regulator of overall lipid metabolism, on one hand by directly stimulating lipid accumulation on WAT reservoirs, and on the other hand by limiting lipid oxidation in the BAT (only via SNS inhibition), liver, and muscle, or indirectly by inhibiting lipolysis.

The increased NPY levels in obesity will predispose to higher stimulation of feeding and lower energy expenditure by decreasing thermogenesis, WAT browning, WAT lipolysis rate, and hepatic β-oxidation while enhancing adipogenesis. All these changes contribute to WAT expansion through physiological lipid accumulation. However, it is not clear how are they involved in or also affected during adipose tissue dysfunction in obesity, thus contributing to the impairments of energy homeostasis and metabolic profile. Given the role of NPY in regulating both arms of the energy balance, it would be an interesting approach to develop a new treatment for obesity mediated by this powerful peptide, although the alterations on the NPY system are still to be fully addressed.

## 3. Mitochondria as a Regulator of Energy Balance and Metabolism

Mitochondria are an extremely important organelle to the function of the cell. Besides regulating cellular bioenergetics, mitochondria are also known to actively participate in processes such as cell death, autophagy, and regulation of immune response, among others [102]. Mitochondria present a well-defined morphology being formed by two different membranes with different compositions and permeability. The OMM is essential for the signaling function since the proteins on this membrane can interact with several cytosolic proteins and establish interactions with other organelles, such as the endoplasmic reticulum (ER) [102]. The inner mitochondrial membrane (IMM) accommodates the respiratory chain [102]. The reducing equivalents produced in the Krebs cycle such as 1,4-dihydronicotinamide adenine dinucleotide (NADH) and dihydroflavine-adenine dinucleotide (FADH_2_), provide electrons to the respiratory chain, which are transported along the different complexes (I–IV). This allows the protons to cross from the mitochondrial matrix to the intermembrane space, generating a proton-motive force used to synthesize ATP. The electrons in this chain are eventually transferred to oxygen in complex IV, producing H_2_O in a four-electron transfer. However, a minor part of the oxygen is only partially reduced, resulting in the formation of ROS [103]. Although ROS are necessary for signaling, excessive production of these products leads to oxidative stress. Mitochondria can change their shape, size, organization, and dynamic interactions, according to cell stimulus. Fusion and fission are the main processes of mitochondria dynamics and are fundamental for normal mitochondria function [104]. Fusion is controlled by two mitofusins: mitofusin 1 (MFN1) and mitofusin 2 (MFN2), which are both present in the OMM, and by optic atrophy protein 1, (OPA1) in the IMM [105]. On the other side, fission is carried out, primarily, by dynamin-related protein 1 (Drp1) [104]. Fusion and more elongated mitochondria are bioenergetically more efficient, being associated with periods of nutritional abundance. During starvation, there is AMPK-driven facilitation of fission (Figure 1B) [102], required for the elimination of dysfunctional mitochondria, that is, mitophagy [102]. Mitochondrial dysfunction, characterized by impaired mitochondrial dynamics and less ATP production, has a major influence on cellular integrity and bioenergetics, constituting a fragility for neuronal circuits, which are highly dependent on ATP [87].

### 3.1. Mitochondria’s Role in Central Control of Energy Balance

Recently, mitochondrial dynamics have been highlighted as an important player in feeding and energy expenditure regulation by the hypothalamus, specifically in NPY/AgRP and POMC neuron’s function. It was demonstrated that mice following food deprivation for 24 h have alterations in mitochondria of NPY/AgRP neurons, inducing higher mitochondrial density, but not greater size. [106]. Following these results, Sungho Jin et al. also described a more fission-like phenotype in mitochondria of 16-h fasted NPY/AgRP neurons [107]. In *Drp1* knockout mice, with consequent disruption of fission, a decrease in mitochondrial respiration was observed, culminating in a lower neuronal firing frequency and in decreased body weight and feeding, together with a failure in ghrelin’s ability to induce hyperphagia [107] Overall, these data suggest that mitochondrial fission is required for the firing of orexigenic neurons during fasting conditions (Figure 1B). Positive energy balance conditions induce a change to a more fusion-like phenotype that is correlated with a lower NPY/AgRP neuron activity [106]. However, in sustained obesogenic conditions, the mitochondrial dynamic seems to alter. Indeed, a 15-week-long HF diet induced mitochondria elongation and a decrease in mitochondrial density, which characterizes fusion processes [106].

Regarding POMC neurons, similar changes and adaptations to metabolic alterations have also been observed, since mitochondria are larger in fed conditions compared with the fasting period (Figure 1C). This is supported by a decrease in DRP1 activation during feeding, indicating that more elongated mitochondria are present in positive energy balance states [108]. Santoro et al. showed that mice with selective deletion of *Drp1* in POMC neurons, presented greater mitochondrial size in these neurons 1 h after glucose administration, accompanied by higher neuronal activity compared with control animals. This suggests that mitochondrial fusion in POMC neurons contributes to increased firing frequency (Figure 1C) [108]. Together, these data display the important dual role played by mitochondrial dynamics in NPY/AgRP and POMC neuron function depending on the fast-feeding cycle. A positive energy state or fed conditions is correlated with a tendency to fusion processes and leads to increased activity of POMC neurons and silencing of the NPY/AgRP neurons [106]. In contrast, in a negative energy state or fasting conditions, a more fission-like phenotype and an increased activity of NPY/AgRP neurons over POMC neurons are observed (Figure 1C) [106]. However, the study of hypothalamic mitochondrial dynamics carried out with protocols resembling more physiologic conditions, that is, responses to shorter fasting and acute meal ingestions are needed to understand the physiology of mitochondria-dependent NPY/AgRP and POMC dual activity

Mitochondrial activity-derived ROS is an important regulator of the firing of both orexigenic and anorexigenic neurons [109]. Andrews et al. suggested that NPY/AgRP and POMC neurons were activated by the modulation of ROS levels in a distinct manner: a decrease in ROS levels activated NPY/AgRP neurons, while POMC neurons activation was dependent on an increase in ROS levels [110]. Indeed, fasted-dependent activation of NPY/AgRP neurons and inactivation of POMC neurons correlates with a decrease in ROS production via an uncoupling protein 2 (UCP2)-dependent manner [110,111]. This uncoupling protein is highly expressed in hypothalamic mitochondria, promoting proton leak from the matrix to the inter-membrane space, leading to the disruption of electron transport chain activity, and thus, lowering ROS production. During fasting periods, UCP2 levels increase in the hypothalamus [111], reducing ROS production, and stimulating the firing of orexigenic neurons (Figure 1B). Corroborating this evidence, the *Ucp2* knockdown in mice increased POMC neurons firing, due to increased ROS production [112]. The mechanism underlying the firing of NPY/AgRP neurons by low levels of ROS is still unknown; however, in POMC neurons, an increase in ROS levels changes the membrane potential, promoting firing [113]. PPAR-γ seems to play an important role in the ROS level regulation in POMC neurons by promoting peroxisome proliferation. PPAR-γ activation occurs when ROS levels increase, and the activation in these neurons functions as a contradictory mechanism to decrease these levels [113]. Indeed, treatment with GW9662, a PPAR-γ antagonist, in DIO mice result in a significant decrease of the number of peroxisomes and an increase of ROS levels in POMC neurons. Consequently, a decrease was observed in the percentage of silent POMC neurons and an increase in the firing frequency, showing that PPAR-γ has an important role in the control of POMC neurons activity through the control of ROS levels [113]. The mitochondrial dimension is proportional to its activity and therefore to ROS production. Thus, in feeding periods, the fusion process is more active, and the ROS levels increase, resulting in a stimulation of POMC neurons (Figure 1C). Corroborating this reasoning, Santoro et al. demonstrated, through deletion of *Drp1* in POMC neurons, that a lower fission process and consequently mitochondrial division leads to greater neuronal activity, which was correlated with an increase in ROS [108]. Regarding energy expenditure, these neurons are known to participate in the control of processes such as thermogenesis in BAT. With this in mind, one can hypothesize that the disruption of mitochondrial function and dynamics in POMC neurons can affect these processes. Indeed, Schneeberger et al. demonstrated that the disruption of fusion mechanisms in POMC neurons affected thermogenesis in BAT. *Mfn2*-specific knockout in POMC neurons caused a decrease in interscapular temperature adjacent to BAT in mutant mice accompanied by decreased *Ucp1* expression [114]. So, mitochondrial dynamics seem to have an important role in the hypothalamic control of the energy expenditure processes; however, more studies are needed to confirm this hypothesis and the mechanisms involved.

The role of ghrelin in activating NPY/AgRP neurons is well established, as already described. ROS signaling plays a determinant role in ghrelin-induced NPY/AgRP neurons firing (Figure 1B). Indeed, intraperitoneal injection of ghrelin results in increased mitochondrial respiration in NPY/AgRP neurons. An augment in ROS levels should be expected; however, a decrease was observed, due to ghrelin-induced upregulation of *Ucp2*. Ultimately, this resulted in an increased firing of NPY/AgRP neurons and food intake in wild-type mice compared with *Ucp2*^–/–^ mice [110]. Further, an increase in mitochondria number during fasting was also observed, suggesting a possible involvement of ghrelin in the regulation of mitochondrial dynamics [110]. On the other hand, leptin appears to be responsible for the increase in ROS levels in POMC neurons upon their activation (Figure 1C). Diano et al. found a positive correlation between leptin, ROS levels, and POMC neuronal activity and, therefore, the stimulation of satiety. They also demonstrated that POMC neurons of *ob*/*ob* mice in fed conditions, with deficient leptin production and impaired neuronal activity, possessed similar ROS levels to wild-type mice in fasted conditions. So, in POMC neurons of *ob*/*ob* mice feeding does not seem to lead to an increase in ROS levels. Therefore, leptin apparently have an important role in the generation of the peak of ROS necessary to the activation of POMC neurons, since disruption of leptin activity, as it happens in *ob*/*ob* mice, impairs ROS production in these neurons. However, 48 h leptin treatment in *ob*/*ob* mice resulted in an increase in ROS production to levels identical to fed wild-type mice [113]. Altogether these studies point to the important role of mitochondria as metabolic mediators.

### 3.2. Obesogenic Diets-Induced Mitochondrial Alterations on Feeding Circuitry

Obesity is associated with neuroendocrine alterations and impaired gut–brain–adipose tissue axis communication. An impairment in nutrient and hormonal sensing and consequent miscommunication to brain centers and peripheral effector organs leads to the inability to maintain energy homeostasis in patients with obesity [109]. Although fundamental to driving an appropriate response during the fed–fasting cycle, the increase of ROS production, typical in metabolic comorbidities due to oxidative stress and several inflammatory processes, can contribute to cell dysfunction [115]. Indeed, HF diet-fed mice present higher levels of ROS in the hypothalamus [116]. Pathological ROS production in the hypothalamus causes oxidative stress in NPY/AgRP and POMC neurons, which can lead to defective nutrient sensing and energy balance regulation. Since HF diets increase ROS levels, it was expected that mice fed with HF diet presented higher POMC neurons activity upon an increase in glucose levels. However, HF diets impair glucose sensing, possibly due to the oxidative stress and inflammation effects, which ultimately can result in the development of insulin resistance in hypothalamic neurons, mainly in POMC neurons, since they are glucose-excitable neurons, resulting in a decrease of neuronal activity. An increase in UCP2 in POMC neurons can also be an explanation for this impaired neuronal function [112,117]. Contributing to the drop in POMC neurons activity, HF diets also induce an increase in PPAR-γ levels [113]. In contrasting to what should be expected, NPY/AgRP neurons are highly activated in HFD-fed mice, increasing inhibitory tone upon POMC neurons [118]. NPY/AgRP neurons seem to change from glucose-inhibited cells to glucose-excitable cells when receiving HFD stimulation, and mitochondria function appears to be important in this process since the deletion of either Mfn1 or Mfn2 impairs neuronal firing in HFD-mice [106]. Therefore, obesogenic diets increase physiologic ROS levels to pathologic levels in the hypothalamus impairing both NPY/AgRP and POMC neuronal function and firing, which leads to a weaker satiety response and a lower energy expenditure.

HF diets promote the downregulation of Mfn2 in the hypothalamus [109], which contributes to an increase in food intake and body weight, suggesting an impairment in ARC neuron’s function. This disruption also affects energy expenditure processes. Schneeberger et al. demonstrated a clear correlation between Mfn2 and the thermogenic process in the BAT. A short-term 4-day of HF diet was sufficient to observe the downregulation of hypothalamic Mfn2 expression. However, the overexpression of this gene in the hypothalamus of DIO mice led to a significant increase in interscapular surface temperature adjacent to BAT and in UCP1 levels in BAT, which points to an enhanced thermogenic process [114]. So, obesity-induced downregulation of Mfn2 can be an important feature of the dysregulation of thermogenesis. Nevertheless, MFN2 also controls mitochondria–ER interactions, which are essential for Ca^2+^ homeostasis. A decrease in MFN2 protein impairs interactions between ER and mitochondria, increasing not only Ca^2+^ levels but also ROS production due to ER stress [109]. Ca^2+^ is a well-known secondary messenger that promotes K^+^ channels activation in neurons promoting changes in their membrane potential and, ultimately, in their activation frequency [119]. Thus, by decreasing MFN2 levels, HF diets lead to oxidative and ER stress, impairing glucose sensing and ARC neurons activity due to the lower fusion rate and Ca^2+^ action on membrane potential.

In the hypothalamus, obesity impairs the sensing of both nutrients and hormones, and changes in mitochondrial function/dynamics and ROS production are apparently crucial factors in compromising the feeding and energy expenditure circuitry and affecting whole-body metabolism and energy balance. Having this in mind, the development of new therapeutics that target mitochondrial dynamics and ROS production modulation in the hypothalamus could be promising for the treatment of obesity and other metabolic disorders related to poor energy balance control.

## 4. NPY as a Regulator of Mitochondrial (dys) Function

The first implications of NPY in regulating mitochondrial activity arose from the study of Kaipio et al. showing the existence of a putative NPY mitochondrial fragment resulting from an alternative translation initiation site [120]. The preproNPY consists of a 97 amino acid sequence containing two equally strong Kozak sequences that can initiate translation [121]. While Kozak-1 gives rise to mature NPY_1-36_, Kozak-2 originates a truncated form, the NPY_17-36_ [121]. N-terminal truncated NPY co-localized within mitochondria [121] and GFP-mediated assays showed that a construct enabling Kozak-2, colocalized exclusively within mitochondria, whereas the construct with both Kozak-1 and mutated Kozak-2 sequence was located in Golgi/ER and showed no mitochondrial targeting [121]. Interestingly, neuronal cells mostly adopt Kozak-1-dependent translation, thus favoring the formation of NPY_1-36_, required for neurotransmission [120]. On the contrary, ovarian and human umbilical vein endothelial cells (HUVEC) displayed substantial translation initiation at Kozak-2, therefore producing relevant amounts of both NPY_1-36_ and the mitochondrial NPY_17-36_ [120]. The differential translational choice, dependent on the cellular type, determines distinct conformations and fate (secretory vesicles or mitochondria) of NPY and may be associated with different functions, according to the needs of a specific cell. However, the precise role of the NPY_17-36_ on mitochondria is yet to be established. The leucine 7 to proline (L7P) polymorphism, originally found by Karvonen et al. in northern European populations [122], consists in the substitution of a leucine residue at position 7 by a proline amino acid and results in an alteration on the tertiary conformation of the NPY protein [123]. The L7P polymorphism regulates the intracellular processing and packaging of the newly synthesized preproNPY [124], resulting in the formation of more mature NPY_1-36_ in cells from the L7P genotype [124]. Further, the L7P polymorphism was suggested to affect preproNPY intracellular mobility. However, GFP-labelled constructs with L7P and L7L genotypes showed similar mobility for the mature and mitochondrial NPY [121], leaving doubts on whether the polymorphism affects preproNPY translocation and mitochondrial NPY formation.

Alterations in NPY conformation and levels are linked to adiposity and altered metabolic control, with circulating NPY being increased in patients with obesity [1]. Accordingly, individuals with the L7P polymorphism have, in addition to higher plasma NPY levels, a higher risk of being overweight or developing obesity and increased serum triglycerides [125]. Several studies in animal models have shown the effects of both centrally and peripherally administered NPY on inducing adiposity, as covered in Section 2.2 [19,97]. However, as an important factor regulating energy homeostasis, NPY has been as well investigated in several models for its ability to modulate mitochondrial activity and thus, the rate of energy production. In fact, in quail myoblasts (QM7 cell line), 100 nM NPY augmented the expression of genes regulating mitochondrial dynamics and biogenesis, while reducing the mitochondrial respiratory rate and ATP production [126]. Since protein levels were not assayed, possible post-translational modifications might account for reduced activation of the mitochondrial machinery, therefore resulting in reduced ATP synthesis. NPY suppressed ATP production in neonatal rat cardiomyocytes since concentrations ranging from 10 to 100 nM of NPY abrogated cellular ATP content [127]. Moreover, 100 nM NPY induced a marked reduction in membrane potential, deformations on cristae and membrane structures, and abnormalities in mitochondria size and shape [127]. In brown adipocytes, 1 nM NPY curbed the adrenergic stimulatory effects on *Ucp1* and *Pgc-1α,* and on genes encoding for the OXPHOS complexes (subunits Vb and VIIc, cytochrome c1, NADH dehydrogenase Fe-S protein 1 and the ATP synthase F1/F0 complex), while having no effect on basal conditions by itself (Figure 1A) [128]. While the mitochondrial biogenesis markers, nuclear respiratory factor 1a *(Nrf1*), and mitochondrial transcription factor A (*Tfam*) were unchanged, mitochondrial activity, including basal respiration and respiratory capacity, was hampered by 1 nM NPY administration even after β3-adrenergic stimulation [128]. Despite all NPY receptors being involved in this response, antagonism of the NPY5R showed the most prominent effects on restoring brown fat markers upon β3-adrenergic agonism plus NPY co-application [128]. These results suggest that peripheral BAT NPY signaling contradicts the SNS-mediated adrenergic effect on inducing browning and OXPHOS. Intraperitoneal NPY injection (120 or 240 ng/100g body weight), in chicken, decreased the expression of genes involved in mitochondrial dynamics (*Mfn2* and *Opa1*) in the breast muscle, while in the leg muscle, NPY induced downregulation of genes involved in mitochondrial biogenesis (*Ppar-γ* and *Pgc1α*) [126]. Peripheral *Npy1r* abolition in mice, resulted in overall increased lipid oxidation, by enhancing CPT-1 levels and proteins of the OXPHOS complexes, as well as β-hydroxy acid dehydrogenase activity, in both liver and muscle, demonstrating the role of NPY1R in blocking mitochondrial activity [98]. Both central and peripheral *Npy1r* knockdown resulted in similar mitochondrial outcomes; however, energy expenditure (kcal heat produced) remained unaltered [98], suggesting that the central NPY-mediated inhibition of thermogenesis is probably not mediated via the NPY1R activation. However, the involvement of NPY receptors and subsequent mechanisms by which NPY may regulate mitochondrial function is still uncertain. While the biological relevance of the putative mitochondrial NPY, (NPY_17-36_)_,_ still needs to be unraveled, it is now clear that the NPY_1-36_, despite not having mitochondrial localization, can regulate mitochondrial function, and thus the energy balance of the cell. Overall, the few studies evaluating NPY effects on mitochondrial performance and dynamics support NPY as a suppressor of mitochondrial activity, thus being able to cease energy production, which is consistent with its lipogenic and anti-lipolytic effects in white adipose depots. However, some studies reported that NPY may increase genes involved in mitochondrial biogenesis [126], raising the hypothesis that it can act in periods of fasting to prepare the cells to respond to nutritional fluxes and enhance energy expenditure in periods of nutritional abundance.

## 5. Concluding Remarks and Future Perspectives

NPY is considered the most powerful factor in regulating energy balance, not only by regulating feeding but also for its crucial role in energy storage and expenditure. Indeed, NPY has a strong relationship with mitochondria. It is possible to describe this connection as a cycle beginning in the brain. During fasting, mitochondrial dynamics in the hypothalamus are altered, promoting mitochondrial fission, a process associated with the firing of orexigenic neurons and the inhibition of POMC neurons, and relying on UCP2-dependent alteration in ROS production [106,107,110,111,112]. Upon NPY release, an orchestrated response driven by the central and peripheral SNS and circulating NPY is initiated to adjust the function of peripheral effector organs according to the energetic status. In several tissues such as the liver, WAT, BAT, and muscle, NPY regulates the expression of genes related to lipid metabolism and mitochondrial function [81,97,98,99,126]. The outcome of the NPY effect on the mitochondria of these tissues leads to a metabolic adaptation converging to lower energy expenditure, by decreasing BAT thermogenesis, WAT lipolysis, and hepatic β-oxidation. These effects may represent NPY-induced energy saving during times of energy shortage.

Obesity is characterized by higher serum and hypothalamic levels of NPY [10,11,129], which in turn are linked to oxidative stress at the central level and compromised neuronal regulation of feeding and energy expenditure. HF diets are associated with impaired mitochondrial dynamics, leading to a higher activity of NPY/AgRP neurons in detriment to anorexigenic signals, potentiating the stimulation of food intake [109]. Hence, NPY levels are increased, compromising ATP production, and contributing to weight gain.

Given that obesity continues to escalate over the years and considering the obesogenic effect of NPY and its role in mitochondrial function, it would be interesting to develop pharmacological strategies for obesity through modulation of the NPY system. Some studies have been reported using NPYRs agonists and antagonists as anti-obesity drugs. However, caution must be used given the ambiguity of the functions regulated by the NPY system, as it may cause secondary effects at neurological levels regarding learning and memory [130]. Indeed, an antagonist of NPY1R (BIBP3226) besides decreasing food intake, alters behavior [89]. Another NPY1R antagonist (J-104870) was characterized by anti-obesogenic effects, decreasing body weight, food intake, and adipocyte hypertrophy; however, it can cross the BBB and cause deleterious effects on behavior and anxiety [131]. Several studies of NPY5R antagonist (MK-0557) demonstrated that body weight is affected in the short term, but as an anti-obesity drug, the effect after 1 year of treatment in humans was nearly absent [89,132]. These studies suggest that the modulation of NPY5R alone may be not enough [132]. The modulation of NPY2R and NPY4R has been poorly described [133]. Regarding NPY2R, the studies reported the icv of NPY_13-36_ agonist induces anxiogenic effects [133,134,135]. However, the agonism of NPY4R is more appealing since peripherally NPY4R activation induces satiety via vagus afferent nerve fibers. Nevertheless, it is suggested that NPY4R modulation appears to be insufficient, requiring a combination with another drug [133]. So, the modulation of the peripheral NPY system arises as a more effective and secure therapeutic approach. The involvement of the NPY1R in the regulation of adiposity suggests a relevant role for a peripheral antagonist to diminish body weight, by decreasing the inhibition of lipolysis coupled to β-oxidation and preventing the over-stimulation of adipogenic/lipogenic processes.

NPY exerts effects in other peripheral tissues, for instance in bone, although the use of NPY1R antagonists may be beneficial to bone mass integrity. NPY1R activation has a negative role in bone mass maintenance, decreasing osteoblast and osteoclast activity, even being involved in osteoporosis development [136].

Also, modulation of the peripheral NPY system allows the control of mitochondria function and dynamics, organelles that are based on oxidative stress and cell metabolism dysregulation. Furthermore, since oxidative stress is important in the pathophysiology of obesity [137], it would be interesting to understand how a combination therapy targeting simultaneously the NPY system and the redox balance with antioxidants could prevent insulin resistance and promote a synergistic effect on genes regulating lipid homeostasis [138]. NPYR-regulating drugs could become a therapy for obesity, regulating energy expenditure on WAT, decreasing body weight and adiposity, and therefore decreasing the predisposition of obesity-associated metabolic diseases.

## Figures and Tables

**Figure 1 biomedicines-11-00446-f001:**
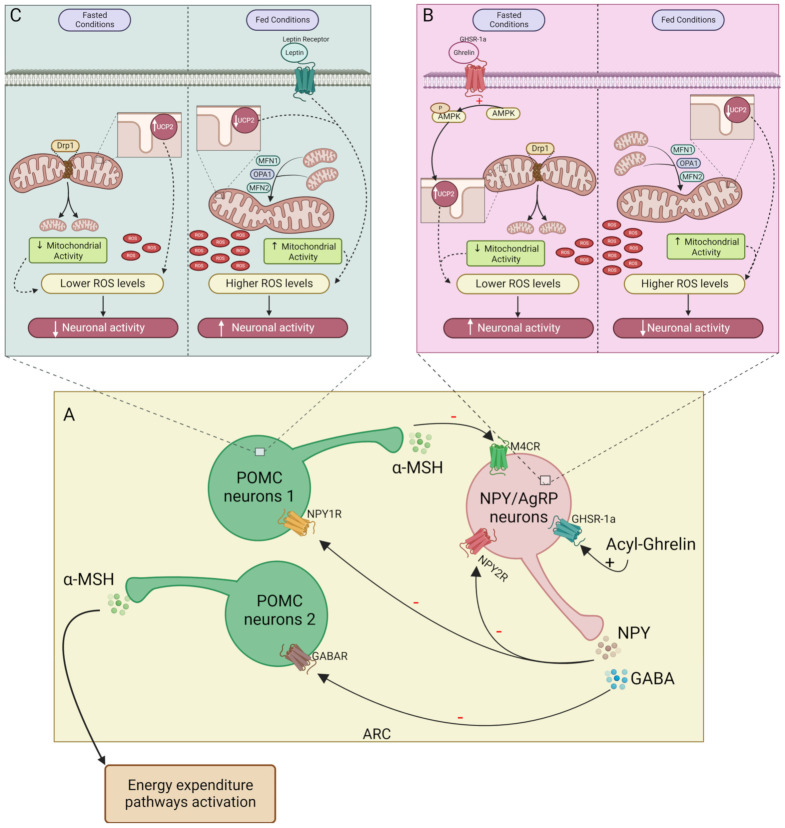
Mitochondrial function on NPY/AgRP and POMC neurons activity. (**A**)—In the ARC during fasting acyl-ghrelin stimulates orexigenic neurons firing, releasing NPY and GABA, inhibiting POMC neurons (subpopulations 1 and 2, respectively). On post-prandial state, α-MSH released by subpopulation 1 of POMC neurons and NPY2R activation inhibits NPY/AgRP neurons; (**B**)—During fasting, ghrelin promotes AMPK phosphorylation in NPY/AgRP neurons. Mitochondrial fission is stimulated by Drp1 and AMPK-induced UCP2 expression contributes to lowering ROS levels. Upon feeding, MFN1, MFN2, and OPA1 promote mitochondrial fusion. The increase in the mitochondrial activity together with a decrease in UCP2 leads to higher ROS levels which inhibit NPY/AgRP neurons activity; (**C**)—On POMC neurons, leptin and mitochondrial activity increased during feeding and contribute to the rise of ROS levels, stimulating POMC neurons activity. In fasting conditions, the ROS levels decrease due to lower mitochondrial activity and increased UCP2, suppressing POMC neuron firing. – inhibitory action; + stimulatory action. NPY/AgRP, neuropeptide Y/Agouti-related protein; POMC, proopiomelanocortin; ARC, arcuate nucleus of the hypothalamus; α-MSH, α-melanocyte-stimulating hormone; NPY2R, NPY receptor 2; GABA, gamma aminobutyric acid; AMPK, AMP-activated protein kinase; Dr1, dynamin-related protein 1; UCP2, uncoupling protein 2; ROS, reactive oxygen species; MFN1, mitofusin 1; MFN2, mitofusin 2; OPA1, optic atrophy protein 1.

**Figure 2 biomedicines-11-00446-f002:**
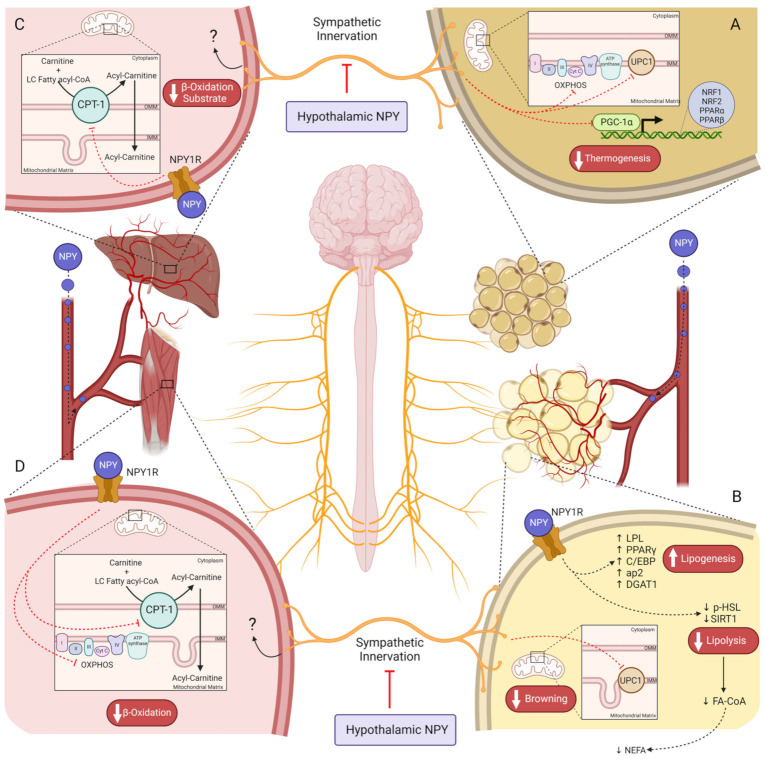
NPY controls energy expenditure by regulating cell metabolism on several tissues. (**A**)—Hypothalamic NPY inhibits thermogenesis in BAT by peripheral SNS suppression, reducing UCP1 levels. NPY downregulates PCG1-a and genes encoding OXPHOS complexes, via sympathetic innervation, reducing β-oxidation and ATP production on brown adipose tissue; (**B**)—NPY in circulation binds to its receptors on adipocytes, increasing adipogenic proteins such as PPARγ, C/EBP, ap2, and DGAT1. Lipolysis is diminished by NPY’s direct action on WAT, decreasing SIRT1 and p-HSL (ser563), and reducing the availability of NEFA for hepatic lipid oxidation. Peripheral sympathetic innervation inhibition by hypothalamic NPY blocks UCP1-mediated WAT browning; (**C**)—In the liver, the substrate for β-oxidation is reduced in consequence of CTP-1 inhibition by NPY1R on hepatocytes. NPY, neuropeptide Y; BAT, brown adipose tissue; SNS, sympathetic nervous system; (**D**)—In skeletal muscle, β-oxidation rate is reduced by NPY1R-induced inhibition of CPT-1 and OXPHOS; UCP1, uncoupling protein 1; PGC1-a, peroxisome proliferator-activated receptor-γ coactivator; OXPHOS, oxidative phosphorylation; PPARγ, peroxisome proliferator-activated receptor γ; C/EBP, CCAAT-enhancer-binding proteins; ap2, adipocyte protein 2; DGAT1, diacylglycerol O-acyltransferase 1; WAT, white adipose tissue; SIRT1, sirtuin 1; p-HSL, p-hormone-sensitive lipase; NEFA, non-esterified fatty acids; CPT-1, carnitine palmitoyltransferase I; NPY1R, NPY receptor 1.

**Table 1 biomedicines-11-00446-t001:** Summary table of NPY receptors-mediated actions according to localization: Hypothalamic and in white adipose tissue.

Receptor	Localization	Outcome
NPY1R	Hypothalamus	↑ Feeding
	WAT	↑ Lipogenesis ↓ Lipolysis
NPY2R	Hypothalamus	↓ NPY release
	WAT	↑ Adipogenesis ↑ Angiogenesis
NPY5R	Hypothalamus	↑ Feeding
	Hypothalamus/WAT?	↑ Adiposity

## Data Availability

Not applicable.

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
