# Peer review of "The Bidirectional Relationship of NPY and Mitochondria in Energy Balance Regulation"

_biomedicines, 2023, doi:10.3390/biomedicines11020446_

Round 1

Reviewer 1 Report

This review focuses on the interconnection between mitochondrial function and dynamics with central and peripheral neuropeptide Y actions, discussing possible therapeutical modulations of the neuropeptide Y system as an anti-obesity tool.

My main concern is that the title does not reflect the manuscript content. In any section the authors describe the effect of NPY in brain mitochondria. There are a lots of studies reporting the role of NPY in the peripheral tissues, but no one in CNS. The authors described the role of mitochondria in NPY/AgRP neuronal function and activation, but this is not the effect of NPY by itself. The title and the purpose of the manuscript should be revised.

Page 1 – Redundant phrases:  In particular, the neuropeptide Y (NPY), is a potent orexigenic peptide pointed out as an obesogenic factor. The NPY system dysfunction is highly associated with obesity. “

Page 1 – Redundant phrases:  “higher levels of NPY in patients with obesity were suggested to promote higher food intake and lower energy expenditure [1]. Moreover, exposure to high-fat (HF) and high-sugar diets in rodents increases NPY levels and sensitivity in the hypothalamus, contributing to weight gain and fat accumulation [2]”.

Page 3: NPY/AgRP neurons are activated by ghrelin in fasting condition (during the in catabolic state), and under fed condition (anabolic state) NPY secretion is inhibited. How can you state that the NPY has mainly in anabolic functions? NPY reduces energy expenditure, and for that it mainly inhibited anabolism.

- Please use different sections to describe the central vs peripheral effect of NPY. The present form of the manuscript makes very difficult to read and understand the different roles.

- Please indicate what stimuli is necessary to produce NPY in the periphery.

- Throughout the manuscript there are a lot of redundancies. Ex in  Page 10, last phrase- “In contrast, in a negative energy state or fasting conditions, is  observed a more fission-like phenotype and an increased activity of NPY/AgRP neurons over POMC neurons.” Similar idea of beginning of the section 3.1 “Food deprivation during 24 hours in mice induced higher mitochondria density in NPY/AgRP neurons, but not greater size, suggesting a tendency to fission processes in these, but not in POMC neurons.”

Another example in section 4: “Studies have been suggesting that the NPY17-36 fragment is

exclusively located in mitochondria.” This sentence is followed by another sentence with exactly the same idea: “ N-terminal truncated NPY co-localized within mitochondria [122] and GFP-mediated assays showed that a construct enabling Kozak-2, colocalized exclusively within mitochondria, while the construct with both Kozak-1 and mutated Kozak-2 sequence was located in Golgi/ER, showing no mitochondrial targeting [122]”.

- The authors described that the NPY system is formed by 3 native forms of peptides. What are the main forms in AgRP neurons? And considering that NPY17-36 fragment is exclusively located in mitochondria, there are any study demonstrating the presence of this fragment in the mitochondria of AgRP neurons? And this could have any role in the mitochondrial fission?

Author Response

Dear Editors,

We thank the reviewers for the time invested in the revision process and for the constructive comments that resulted in improvements in the manuscript. In this document, we respond, point by point, to the concerns and questions raised. The alterations made in the manuscript following the reviewers’ questions are in track changes mode in the text (and detailed in each reply) so that they can be easily found.

We believe we have made all the required alterations and hope that they meet your expectations.

Response to Reviewer 1 Comments

Point 1: My main concern is that the title does not reflect the manuscript content. In any section the authors describe the effect of NPY in brain mitochondria. There are a lots of studies reporting the role of NPY in the peripheral tissues, but no one in CNS. The authors described the role of mitochondria in NPY/AgRP neuronal function and activation, but this is not the effect of NPY by itself. The title and the purpose of the manuscript should be revised.

Response 1: The reviewer raised a pertinent point. The title was changed to better fit the content of the manuscript.

Point 2: Page 1 – Redundant phrases: In particular, the neuropeptide Y(NPY), is a potent orexigenic peptide pointed out as an obesogenic factor. The NPY system dysfunction is highly associated with obesity.

Response 2: We acknowledge that these two sentences were redundant, therefore, we eliminated the second phase (lines 38 and 39).

Point 3: Page 1 – Redundant phrases: “higher levels of NPY in patients with obesity were suggested to promote higher food intake and lower energy expenditure [1]. Moreover, exposure to high-fat (HF) and high-sugar diets in rodents increases NPY levels and sensitivity in the hypothalamus, contributing to weight gain and fat accumulation [2]”.

Response 3: We acknowledge that these two phrases may be confusing, however, we intended to convey two messages: 1) patients with obesity are characterized by higher levels of NPY, promoting an energy imbalance; 2) a hypercaloric diet in rodents, possibly by potentiating NPY neurons activity, increases NPY levels, suggesting that changes in NPY levels may precede obesity.  We have rewritten it to improve clarity and avoid misleading interpretation, thank you for your suggestion (lines 39 to 43).

Point 4: Page 3: NPY/AgRP neurons are activated by ghrelin in fasting condition (during the in catabolic state), and under fed condition (anabolic state) NPY secretion is inhibited. How can you state that the NPY has mainly in anabolic functions? NPY reduces energy expenditure, and for that it mainly inhibited anabolism.

Response 4: This must be pictured in a multidimensional way, of time (post-absorptive vs post-prandial) and place (central vs at the periphery). During fasting, NPY is released from AgRP/NPY neurons to induce food intake (lines 87-96) and reduced energy expenditure (decreasing catabolism). Then, central NPY is, in fact, inhibited during a positive energy state.

The peripheral effects of NPY are less known as well as the NPY plasma fluctuations upon a meal. We know that: 1) NPY directly stimulates lipogenic programs and inhibits lipolytic processes when administered to adipocytes, having similar effects on the liver and muscle; 2) On the other hand, hypothalamic NPY does not induce energy storage,  only inhibits previously stimulated lipolysis, possibly mediated by NPY5R (lines 265 to 282); 3) insulin induces NPY release from peripheral tissues, such adipose tissue and pancreas (lines 334 and 336), suggesting that NPY promotes anabolic process during postprandial state. 4) NPYRs are coupled to Gi protein, reducing cAMP-dependent kinases, being, therefore, more related to de novo synthesis than to catabolism itself. Nevertheless, more studies are necessary to understand the effects of central NPY on metabolism and also the impact of short and long fasting since intermittent fasting is associated with high energy expenditure.

Point 5: Please use different sections to describe the central vs peripheral effect of NPY. The present form of the manuscript makes it very difficult to read and understand the different roles.

Response 5: Thank you for the advice. We divided section 2.2 into two subsections: 2.2.1 Central NPY effects and 2.2.2 Peripheral NPY effects (lines 242-412).

Point 6: Please indicate what stimuli is necessary to produce NPY in the periphery.

Response 6:  In pancreatic islets, the insulin secretagogue, glibenclamide also leads to NPY release (J Endocrinol. 2000 May;165(2):509-18). Moreover, GLP-1 also stimulates both insulin and NPY release from the pancreas. Thus, as NPY inhibits insulin release, it is suggested as a negative feedback mechanism that NPY release is a response to high insulin levels, preventing excessive levels of insulin (J Endocrinol. 2000 May;165(2):509-18). In adipocytes, treatment with insulin stimulates NPY release (Am J Physiol Endocrinol Metab. 2007 Nov;293(5):E1335-40). So, in the periphery, insulin is responsible for inducing NPY release which may occur during the postprandial state since insulin levels increase upon a meal (lines 332-334).

Point 7: Throughout the manuscript there are a lot of redundancies. Exin Page 10, last phrase- “In contrast, in a negative energy state or fasting conditions, is observed a more fission-like phenotype and an increased activity of NPY/AgRP neurons over POMCneurons.” Similar idea of beginning of the section 3.1 “Food deprivation during 24 hours in mice induced higher mitochondria density in NPY/AgRP neurons, but not greater size, suggesting a tendency to fission processes in these, but not in POMCneurons.”

Response 7: We acknowledge that these two sentences were redundant, we changed the second sentence  (lines 448-452).

Point 8: Another example in section 4: “Studies have been suggesting that the NPY17-36 fragment is

exclusively located in mitochondria.” This sentence is followed by another sentence with exactly the same idea: “ N-terminal truncated NPY co-localized within mitochondria [122] and GFP-mediated assays showed that a construct enabling Kozak-2, colocalized exclusively within mitochondria, while the construct with both Kozak-1 and mutated Kozak-2 sequence was located in Golgi/ER, showing no mitochondrial targeting [122]”.

Response 8: Thank you for calling that to our attention. Indeed, the first sentence is unnecessary and has been removed to improve readability.

Point 9: The authors described that the NPY system is formed by 3 native forms of peptides. What are the main forms in AgRPneurons? And considering that NPY 17-36 fragment is exclusively located in mitochondria, there are any study demonstrating the presence of this fragment in the mitochondria of AgRP neurons?And this could have any role in the mitochondrial fission?

Response 9: As described in the manuscript, NPY is produced in NPY/AgRP neurons, being the main form present in orexigenic neurons (line 107). The other two forms (PYY and PP) are produced in the gut and pancreas (line 230 and 240), and only PYY can act on AgRP neurons, binding to NPY2R. We apologize for the lack of information regarding the production location of PYY. Regarding NPY 17-36, there are no studies demonstrating its presence in the mitochondria of AgRP neurons specifically. The study mentioned in the manuscript is the only one that reported this truncated form of NPY, using the neuronal SK-N-BE cell line.

Reviewer 2 Report

Diana Sousa, and her collaborators, in the paper titled "Mitochondria as a key player in NPY-Mediated energy balance regulation," has tried to provide a new point of view about a possible link between neuropeptide Y and mitochondria.

They propose to act on the functionality of neuropeptide Y in the modulation of mitochondrial function and oxidative stress so that by improving mitochondrial well-being, it may be possible to modulate energy expenditure and reduce metabolic pathologies such as obesity.

In my opinion, they have hit the target in their treatise, explaining it in a very exhaustive way and providing a new interpretation of the treatment of these pathologies.

It could also be interesting to evaluate a combination of treatment with antioxidants since, as demonstrated by Fasciolo G. et al., oxidative stress can also contribute alone to inducing obesity-related pathologies such as insulin resistance.

Author Response

Dear Editors,

We thank the reviewers for the time invested in the revision process and for the constructive comments that resulted in improvements in the manuscript. In this document, we respond, point by point, to the concerns and questions raised. The alterations made in the manuscript following the reviewers’ questions are in track changes mode in the text (and detailed in each reply) so that they can be easily found.

We believe we have made all the required alterations and hope that they meet your expectations.

Response to Reviewer 2 Comments

Point 1: It could also be interesting to evaluate a combination of treatment with antioxidants since, as demonstrated by FascioloG. et al., oxidative stress can also contribute alone to inducing obesity-related pathologies such as insulin resistance.

Response 1: Thanks for your interesting suggestion. We have added a sentence addressing this topic (lines 724 to 727).

Reviewer 3 Report

Neuropeptide Y plays a major role in the control of feeding and energy expenditure. In this review, Sousa et al focus on the interconnection between mitochondrial function and dynamics and the central and peripheral actions of neuropeptide Y. This review is organized in 3 parts: 1) NPY as the master regulator of energy balance; 2) mitochondria as a regulator of energy balance and metabolism; 3) NPY as a regulator of mitochondrial (dys)function.  
This review is well done with good illustrations. The final discussion on possible therapeutic modulations of the neuropeptide Y system as an anti-obesity tool with potential risks is also interesting.

Author Response

Dear Editors,

We thank the reviewers for the time invested in the revision process and for the constructive comments that resulted in improvements in the manuscript. In this document, we respond, point by point, to the concerns and questions raised. The alterations made in the manuscript following the reviewers’ questions are in track changes mode in the text (and detailed in each reply) so that they can be easily found.

We believe we have made all the required alterations and hope that they meet your expectations.

We thank reviewer #3 for the positive comments, we are glad you enjoyed the reading.

Round 2

Reviewer 1 Report

The authors properly addressed all the questions raised. I reccomend the publication of the manuscript.